# Biorecognition Engineering Technologies for Cancer Diagnosis: A Systematic Literature Review of Non-Conventional and Plausible Sensor Development Methods

**DOI:** 10.3390/cancers14081867

**Published:** 2022-04-07

**Authors:** Kalaumari Mayoral-Peña, Omar Israel González Peña, Alexia María Orrantia Clark, Rosario del Carmen Flores-Vallejo, Goldie Oza, Ashutosh Sharma, Marcos De Donato

**Affiliations:** 1School of Engineering and Sciences, Campus Queretaro, Tecnologico de Monterrey, Av. Epigmenio González No. 500, San Pablo, Queretaro 76130, Mexico; kmayoralp@gmail.com (K.M.-P.); asharma@tec.mx (A.S.); 2School of Engineering and Sciences, Campus Monterrey, Tecnologico de Monterrey, Av. Eugenio Garza Sada Sur No. 2501, Tecnológico, Monterrey 64849, Mexico; 3Institute for the Future of Education, Tecnologico de Monterrey, Av. Eugenio Garza Sada Sur No. 2501, Tecnológico, Monterrey 64849, Mexico; 4School of Engineering and Sciences, Campus Mexico City, Tecnologico de Monterrey, C. Puente 222, Ejidos de Huipulco, Tlalpan, Mexico City 14380, Mexico; a01655427@tec.mx; 5Department of Biomedical Engineering and Mechatronics, Campus Toluca, Universidad del Valle de México (UVM), C. De Las Palmas Poniente 439, San Jorge Pueblo Nuevo, Metepec 52164, Mexico; rosariofvallejo@gmail.com; 6Laboratorio Nacional de Micro y Nanofluídica (LABMyN), Centro de Investigación y Desarrollo Tecnológico en Electroquímica (CIDETEQ), Parque San Fandila, Pedro Escobedo, Queretaro 76703, Mexico; goza@cideteq.mx

**Keywords:** biosensor, cancer diagnosis, point-of-care, molecularly imprinted polymers, antibody mimetic molecules, recombinant antibodies, cancer biomarkers, exosomes, aptamer, phage display

## Abstract

**Simple Summary:**

Approximately 70% of patients with cancer are diagnosed at late stages of the disease in developing countries. This is partly owed to the restricted access to cost-effective and accurate diagnostic tools in healthcare systems. Biosensor diagnostic tools based on conventional antibodies have been a valuable option for creating accessible detection systems for cancer. However, antibodies have certain limitations related to cost, stability, and applicability. The latter promoted the research and development of alternative approaches to generating molecules and molecule-based scaffolds with similar biorecognition properties to antibodies (non-conventional technologies). This review aimed to present and analyze the current trends of three of these emerging non-conventional technologies for biorecognition engineering in cancer diagnostics, named: molecularly imprinted polymers, recombinant antibodies, and antibody mimetic molecules. These non-conventional technologies are promising, relevant, and more accessible alternatives to conventional antibodies in developing cancer biosensors and worthy of being acknowledged by the scientific community, especially for their use in point-of-care cancer diagnostics in developing countries.

**Abstract:**

Cancer is the second cause of mortality worldwide. Early diagnosis of this multifactorial disease is challenging, especially in populations with limited access to healthcare services. A vast repertoire of cancer biomarkers has been studied to facilitate early diagnosis; particularly, the use of antibodies against these biomarkers has been of interest to detect them through biorecognition. However, there are certain limitations to this approach. Emerging biorecognition engineering technologies are alternative methods to generate molecules and molecule-based scaffolds with similar properties to those presented by antibodies. Molecularly imprinted polymers, recombinant antibodies, and antibody mimetic molecules are three novel technologies commonly used in scientific studies. This review aimed to present the fundamentals of these technologies and address questions about how they are implemented for cancer detection in recent scientific studies. A systematic analysis of the scientific peer-reviewed literature regarding the use of these technologies on cancer detection was carried out starting from the year 2000 up to 2021 to answer these questions. In total, 131 scientific articles indexed in the Web of Science from the last three years were included in this analysis. The results showed that antibody mimetic molecules technology was the biorecognition technology with the highest number of reports. The most studied cancer types were: multiple, breast, leukemia, colorectal, and lung. Electrochemical and optical detection methods were the most frequently used. Finally, the most analyzed biomarkers and cancer entities in the studies were carcinoembryonic antigen, MCF-7 cells, and exosomes. These technologies are emerging tools with adequate performance for developing biosensors useful in cancer detection, which can be used to improve cancer diagnosis in developing countries.

## 1. Introduction

### 1.1. Glossary and Terminology

Evolutionary processes in cells have led to the development of a diverse repertoire of receptors to which biomolecules can specifically bind. Many biological functions depend on this specific binding called biorecognition, considered to be an essential process in living organisms [1]. Biological systems frequently use proteins (such as enzymes, membrane receptors, and antibodies) for biorecognition because of their chemical structure (shape complementarity with target molecules) [1,2].

Antibodies are highly specific defense molecules, mostly acknowledged to be synthesized as part of the adaptive immune response in complex organisms. In healthy individuals, antibodies play several relevant roles in maintaining homeostasis, such as recognizing exogenous agents (e.g. binding to pathogen’s antigens and neutralizing them). Due to their low dissociation constant (in the order of 10^−7^ to 10^−12^) and the feasibility to produce them on a large scale, antibodies are one of the most common biorecognition elements used for diagnostic technologies and biosensors [3,4].

In this article, the term cancer biomarker is used to describe distinctive, naturally occurring molecules found in biological samples of patients with cancer (e.g. proteins, polysaccharides, nucleic acid sequences, etc.). The term cancer entity refers to supramolecular biological structures found in cancer samples (e.g. exosomes, cells, and organelles, among others). These biomarkers or entities are associated with a particular physiological or pathological process or stage in patients with cancer. On the other hand, the term biorecognition engineering refers to the synthetic development of bioreceptors through different technologies capable of interacting with cancer biomarkers and cancer entities. This capability can be used, for instance, in the construction of diverse technologies such as molecular diagnostic tools [5], therapeutic [6], and even theranostic agents [7]. Biorecognition engineering technologies can overcome some limitations or disadvantages of conventional diagnostic tools based on antibodies, making them a promising alternative.

Three types of emerging biorecognition engineering technologies were analyzed in this article: molecularly imprinted polymers, recombinant antibodies, and antibody mimetic molecules. Each one of these technologies shows distinctive characteristics: (a) molecularly imprinted polymers use the target molecule as a template; (b) recombinant antibodies are produced by recombinant technologies, such as phage display; and (c) antibody mimetic molecules (aptamers, affibodies, and affimers) are engineered molecular structures, different from antibodies, that are designed for specific biorecognition. Thus, these three technologies are useful emerging tools for developing biosensors and are being used in other diagnostic strategies to detect cancer in patients at the point-of-care with high accuracy, minimal invasiveness, and rapid results.

### 1.2. Background

According to the World Health Organization, cancer is the second leading cause of mortality globally, responsible for approximately 10 million deaths a year [8]. Accessibility to cancer diagnostic tools is insufficient in developing countries, which is one of the main factors why 70% of cancer cases are diagnosed at advanced stages. Consequently, the probability of having an effective treatment against this disease is limited, and the probability of survival of patients is low. As a result, developing more accessible and accurate diagnostic tools (as biosensors and medical devices) is necessary, especially in countries with limited resources in the healthcare system.

Biosensors are analytical instruments that monitor the state of a system by measuring signals through a molecular biorecognition element (receptor) in response to its binding to a particular analyte (e.g., cancer biomarkers). Biosensors consist fundamentally of a receptor and an analyte that binds to it, a transductor, an amplifier, and an analytical system. Biosensors transform specific biorecognition events so that they can be processed and quantified to be understood by the user and thus assist in decision making (e.g., clinical diagnosis of cancer patients). The molecular biorecognition element is essential in biosensors and diagnostic device development [9]. The biosensor’s performance is closely related to the biorecognition element’s sensitivity and specificity [10]. For this reason, the selection of the biorecognition element is a task that should be conducted carefully. Biosensors have important advantages over conventional laboratory techniques: portability, low cost, ease of use, and high sensitivity [11].

Many conventional technologies in biosensor development use the immune system of animals (commonly mammals) to produce polyclonal antibodies, which come from different parental cells and bind to multiple epitopes of the target molecule. Others use hybridoma cell lines (fusion of B cells and myeloma cells) to produce monoclonal antibodies (produced by clones from the same parental cells) that bind to a single epitope of the interested molecule [12,13]. However, these conventional technologies have significant disadvantages: the target molecule must be immunogenic, the production yield is low, the use of animals or hybridoma cell lines is required, and the purification of the antibodies is expensive [13]. On the other hand, antibodies are subject to degradation at extreme pH or temperature [14].

Alternative biorecognition technologies have been developed to overcome the limitations of conventional antibody production. Some of the most relevant are molecularly imprinted polymers, recombinant antibodies, and antibody mimetic molecules. There is a fertile field where biorecognition engineering technologies can be implemented [10]. Progress in this field can enable the creation of cheaper and more efficient diagnostic tools for cancer.

### 1.3. Aim of the Study

Although some review articles have previously explained the biorecognition non-conventional technologies and their applications, they were not focused on cancer diagnostics in challenging environments. For this reason, it is relevant to establish if these technologies can be used to develop low-cost and minimally invasive devices for early detection and continuous monitoring of this disease. This systematic literature review was focused on presenting, comparing, and promoting these technologies used in cancer diagnosis in the community of healthcare professionals. The following general objectives were established to accomplish this goal:Objective 1: Explain the fundamentals of the three biorecognition technologies (molecularly imprinted polymers, recombinant antibodies, and antibody mimetic molecules).Objective 2: Establish the advantages and disadvantages of these technologies and their applications and perspectives in cancer diagnosis and therapeutics.Objective 3: Report and analyze the results of the literature search about the current use of these technologies in the development of biosensors for cancer diagnosis.

To accomplish the first and second objectives, a brief explanation and some relevant applications of each technology in cancer diagnosis and therapy are provided in the results section. For the third objective, the discussion section provides a synthesis of the systematic literature review analysis by trying to address the following research questions:RQ1: Is it possible to implement these technologies for the detection of cancer?If the answer is yes, then the following questions can be addressed:RQ2: Which of these three technologies has been more extensively studied?RQ3: What types of cancer have been detected using these three technologies?RQ4: Is it possible to detect different cancer types using a single biomarker or cancer entity?RQ5: What methods are used for the detection of cancer by using these biomarkers or cancer entities?RQ6: Which biomarkers and cancer entities are the most commonly studied?RQ7: What are the cancer detection levels reached using these three technologies?

The second section of this article addresses the methodology used to answer the research questions. The third section describes the three technologies and analyzes their advantages and disadvantages and some applications in cancer diagnostics and treatment. In the fourth section, a discussion is presented regarding the most studied non-conventional technology for cancer detection of the three analyzed, the molecules that have been used, the types of cancer that have been detected, the detection methodologies used, and the study limitations of this review. The fifth section mentions the conclusions, covering the research questions and future perspectives. Finally, a summary of the data extracted from the reviewed literature is provided in a table in Appendix A. The abbreviation list contains all the acronyms used in this article for the reader’s convenience.

## 2. Methodology

### 2.1. Data Sources and Searches

Scientific literature analysis was performed using PRISMA guidelines [15] and the Clarivate Web of Science database (from 2000 up to 10 December 2021) [16]. In this process, it was not necessary to register this study in specialized platforms for an international prospective register of systematic reviews because the information presented did not necessarily correspond to experimental studies in patient samples or have the purpose of observing their trends. Instead, this systematic literature review focused on describing the advantages and disadvantages of the three non-conventional biorecognition engineering technologies for cancer diagnosis according to the current applications, trends, and potential future applications by emphasizing relevant aspects, for instance: the detection methods, cancer limit of detections, the lowest concentration tested, the kinds of cancer that the methods can detect, or if these technologies can detect multiple cancers with a particular biomarker of cancer entities, identification of the biomarker of cancer entities used, etc.

The search strings and filtering steps of this process are presented in Table 1:

### 2.2. Inclusion and Exclusion Criteria

Research articles in English were included. Only peer-reviewed published articles were considered. Reviews, book chapters, proceedings, meeting abstracts, and other documents were excluded. The inclusion and exclusion criteria are presented in Table 2.

### 2.3. Data Management

The authors extracted and combined data from the selected articles following the mentioned considerations and presented it as a flowchart in Figure 1. A total of 131 recent articles that reflect the most current technologies and the most used cancer biomarkers and cancer entities were analyzed in this study. Likewise, these 131 articles addressed the mentioned three biorecognition techniques.

## 3. Results

The biorecognition engineering technologies that are described in this section are molecularly imprinted polymers, recombinant antibodies, and antibody mimetic molecules. A brief conceptual explanation, advantages and disadvantages, and examples of applications for cancer diagnosis and therapy are presented for each technology.

### 3.1. Molecularly Imprinted Polymers

Molecularly imprinted polymers (MIP) is a technology that consists of the construction of ligand-selective recognition sites in synthetic polymers, where a template molecule is used during polymerization or polycondensation [17]. The process to generate molecularly imprinting polymers is described in Figure 2. MIP technology is a promising and low-cost method for creating solid materials that can interact with a specific molecule of interest [18]. These materials are generally stable in different pH and temperature conditions [17]. The advantages and disadvantages of this technology are presented in Table 3.

According to Table 3, the MIP technology has lower production costs, can be manufactured in larger quantities, and has a higher lifespan at room temperature than conventional technologies. However, this new technology has important limitations in binding capacity, non-specific binding, and accessibility to the binding sites. Additionally, rearrangements of the polymer can occur, and it is possible to have unstable three-dimensional conformations [17,18,19,20].

Fluorescent bioimaging of hyaluronic acid (HA) is useful for cancer diagnostics and therapy. Typically, HA is stained in two steps (first with a biotinylated HA binding protein and then with streptavidin-FITC). However, rhodamine-labeled MIP can stain HA in only one step and presents various advantages like physical and chemical stability and size adaptability in addition to being cheaper than conventional biological probes such as hyaluronic acid binding protein (HABP) [21]. This facilitates the detection and quantification of HA, which is useful for monitoring tumor development and other conditions. Another example of MIP consists of silica nanoparticles embedded with gadolinium-doped silicon quantum dots and chlorin E6, which provide a resonance dual-imaging while delivering antitumor drugs for photodynamic treatment for cancer. Peng et al. designed this nanocarrier against CD59 epitope on tumors to detect and kill cancer cells, and they showed great specificity for target cells and improved efficacy for synergistic therapy. This example of MIP did not present damage or toxicity to healthy tissues and organs [22]. The same case exists for human epidermal growth factor receptor-2 (HER2), a protein expressed in several types of cancer, including breast cancer. 

Interestingly, imprinted polymers containing fluorescent biomarkers and doxorubicin are also used as a treatment alternative. Therefore, they can perform cellular fluorescent imaging for diagnosis as well as selectively attacking breast cancer cells [23]. Additionally, the human fibroblast growth-factor-inducible 14 with glucose (Glu-FH) and bleomycin (BLM) have been used as templates so that silicon nanoparticles with drug delivery systems bind to pancreatic BxPC-3 cancer cells [24].

On the other hand, radiation therapy is considered a less invasive cancer treatment. However, large doses may damage healthy tissues and organs. Because of this, gold nanoparticles (Au NPs) have been designed as biocompatible radiation sensitizers. This material is incorporated into MIP microgels, and their effects have been investigated in pancreatic cancer [25]. Finally, MIP technology may also be used for the electroanalysis of different drugs. In another instance, an imprinted polymer was created to target the drug 6-mercaptopurine (6-MP), which is used to treat leukemia. For this, hollow carbon nanospheres were decorated with palladium. Then, nitrogen atoms were introduced to the carbon nanospheres to increase conductivity, catalysis, and stability. It created an easier method for detecting 6-MP in plasma, which is a drug that restricts the production of adenine [26].

MIP is a multifunctional platform that allows the detection and treatment of cancer. These polymers can be devised as nanoparticles that permit the encapsulation of MRI contrast agents, PET agents, antitumor drugs, and different kinds of nanoparticles, especially fluorescent nanoparticles, such as quantum dots. They can be used for specific detection of cancer biomarkers and for delivering different drugs that can induce apoptosis or programmed cell death in cancer cells. Figure 3 depicts the encapsulation of different photosensitizers, fluorescent nanoparticles, antitumor agents for cancer cell tracking, and drug delivery after exposure to the laser.

### 3.2. Recombinant Antibodies (Antibody-Based Molecules)

The technology to create antibody-based molecules relies on the use of recombinant antibodies and protein engineering for the synthesis of antibodies’ or immunoglobulin’s fragments [27]. This technology does not require the use of immunized animals or hybridoma cell lines, which are expensive and have low production efficiency [28]. It allows the improvement of antibody stability, solubility, and specificity. Some examples of this technology include engineered antibody fragments (nanobodies) and recombinant antibodies produced by phage display [29,30]. Phage display is a technology for presenting protein sequences on the surface of lysogenic filamentous bacteriophages, which allow for the creation of libraries (a high number of variants) used for affinity screening [30]. The process to generate recombinant antibodies through phage display is described in Figure 4. Phage M13, shown in this figure, is commonly used in this technology. The advantages and disadvantages of recombinant antibodies are presented in Table 4.

According to Table 4, recombinant antibodies technology has lower production costs, has a better affinity for the analytes that are not immunogenic, and can be implemented in recombinant bacterial reactors with higher yields than conventional technologies. However, its development requires recombinant technology and protein engineering.

Recombinant antibodies improve diagnosis accuracy through biomarker detection. An example can be found in the generation of a phage display library that was screened against recombinant HEK-293 cells with the variant CD44 biomarker for gastric cancer cells. Through immunofluorescence analysis, HSCORE, ELISA assay, and other characterization tests, it was observed that one phage candidate, denominated ELT, was the best one to diagnose gastric cancer via CD44v6 biomarker [31]. On the other hand, single-chain variable-fragments (scFv), which are fusion proteins of the variable regions of the heavy and light chains of an antibody (V_H_ and V_L_) connected with a short linker peptide of 10–25 amino acids, may be coupled with phage display libraries to create recombinant antibodies, as was the case of Tadano et al. [32], where candidate phages were cloned to bind with higher specificity and affinity to a cancer stem-like antigen, resulting in a possible new therapy for patients with carcinoma and bone sarcoma. Regarding breast cancer therapy, the Delta-like ligand 1 (DLL1) in Notch signaling becomes a target for this type of cancer. Sales-Dias et al. [33] developed anti-DLL1 antibodies from scFvs. As a result, the antibody IgG-69 was able to disrupt the DLL1 activation of the Notch signaling pathway.

Finally, the prostate-specific membrane antigen (PSMA) has been widely used to detect prostate cancer, but it is prone to poor cancer diagnosis and treatment. In Rosenfeld et al. [34], mRNA coding for anti-PSMA was isolated from camel serum and used to construct a bacterial expression system employed in a phage display system to produce anti-PSMA nanobodies that could effectively bind to PSMA-expressing cells. This study explored the potential of using nanobodies for diagnosis and therapy for prostate cancer and possibly other types of cancer. Figure 5 illustrates how the phage libraries that possess high affinity towards cancer cells are synthesized and how such phages are fluorescently labeled to detect cancer cells using flow cytometry.

### 3.3. Antibody Mimetic Molecules

Antibody mimetic molecules are highly selective biorecognition elements because they are specifically designed to bind the target molecule. Antibody mimetic molecules are biorecognition agents not directly based on antibodies (immunoglobulins), making them different from the mentioned recombinant technology. They can be made of amino acids, DNA, RNA, or organic molecules. The affinity for the analyte is commonly obtained by design or by directed evolution in vitro [35]. The most used antibody mimetic molecules are aptamers, affimers, and affibodies. Aptamers are DNA or RNA oligonucleotides with high affinity and specificity for a target molecule; they are generated in vitro by a process called SELEX [36]. The process to generate aptamers using the SELEX methodology is described in Figure 6. Affimers are small proteins optimized to bind a target molecule; they are produced by phage display and have a structure consisting of an alpha-helix and anti-parallel beta-sheets [37]. Affibodies are engineered proteins designed to bind a specific ligand; they are based on a three-helix bundle domain [38]. The advantages and disadvantages of antibody mimetic molecules are presented in Table 5.

According to Table 5, the antibody mimetics technology has a better affinity for the analytes, is not immunogenic, can easily functionalize surfaces and biosensors, and has a bigger repertoire of analytes in comparison to conventional technologies. However, the development of this new technology requires library selection and combinatorial mutagenesis, produces weaker signals than antibodies, and the design and selection methods are complex and time-consuming. On the other hand, these technologies are patented, limiting their commercial application to a few enterprises.

An example of antibody mimetic molecules is the BH3-mimetic drugs that have been developed to induce apoptosis in malignant cancer cells. These drugs bind to the BCL-2 proteins, which regulate the survival or apoptosis in cells. Therefore, they inhibit the prosurvival BCL-2 proteins, so cancerous cells may go through apoptosis and provide an alternative for cancer therapy. This inhibitor for BCL-2 is already approved by the US Food and Drug Administration and worldwide for chronic lymphocytic and myeloid leukemia [41]. Radioligand therapy (RLT) is also an option for cancer diagnostics and biosensors. In this type of therapy, a molecule that is able to recognize the antigen is radiolabeled with a radionuclide in order to supply ionization energy so that cancer cells can be eliminated [42]. In recent years, the effect of albumin binders along with RLT have been studied, and it has been observed that they may improve the dose uptake and delivery with potential cost-effectiveness.

Colorimetric detection for cancer diagnosis with nanomaterials instead of conventional methods has also been developed. Some examples of these nanomaterials are the MUC1 aptamers to recognize MUC1 proteins on cell surfaces with silver nanoprisms and gold nanoparticles to detect the prostate-specific antigen for prostate cancer [43]. MUC1 exposes tumor-associated carbohydrate antigens (TACAs) like the Tn antigen (single *N*-acetylgalactosamine α O-linked). This Tn antigen is expressed in various aggressive cancers but is absent in normal cells. The synthesis of 2-deoxy-2-thio-a-O-glycohomoglutamates has been explored as a more stable and improved immunogenicity analog for the Tn antigen in order to improve the diagnostic efficiency for this biomarker [44].

Antibody mimetic molecules have a wide variety of applications. For example, nucleic acid aptamers can bind to different proteins for diagnostic purposes, such as cancer cell-specific membrane receptors with high specificity and binding affinities similar to antibodies (Figure 7A). Affibodies are antibody mimetic molecules that can be conjugated with anticancer drugs in order to release their content in specific cells through their interaction with specific receptors (Figure 7B), reducing the secondary effects due to the incorporation of the drug in non-target cells.

In addition, two-dimensional nanomaterials have also been used in breast cancer detection and treatment due to their added qualities. One example of these 2D nanomaterials is layered double hydroxides (LDHs) and nanoclays. They are great nanovehicles for chemotherapy and theranostics because of their high efficiency in loading drugs, molecules, and nanoparticles. For example, Zheng et al. [45] used MgAl-LDHs to attack drug-resistant breast cancer cells while also delivering nanoparticles with cytotoxic selenite. Nanoclays, just like LDHs, are good for ion exchange capacity and delivery systems for biomolecules and other compounds.

## 4. Discussion

### 4.1. Scientific Literature Analysis

A total of 6586 scientific articles about cancer biosensors and 904 about the application of the three biorecognition engineering technologies for the detection of this disease were indexed in the Web of Science from 2000 to 2021, according to the selection process mentioned in the methodology. The distribution of articles by the publication year of these topics is shown in Figure 8.

The number of articles about cancer biosensors and antibody mimetic molecules had an increasing trend from 2004 to 2018, recombinant antibodies from 2014 to 2017, and molecularly imprinted polymers from 2013 to 2016. Additionally, the number of publications on these topics has been maintained since 2018. On the other hand, the scientific articles about the three biorecognition technologies represented 13.7% of the articles about cancer biosensors, which are times shorter than those about conventional antibodies. It indicates that these technologies are used for cancer detection (RQ1). The most studied biorecognition technology was antibody mimetic molecules (11.4%) (RQ2). This data suggested that these biorecognition methods are relatively new and valuable in cancer detection. However, there is still a large room for improvement to reach maturity and large-scale implementation.

According to the methodology, 131 scientific articles published in the last three years were analyzed to answer the research questions. The results of this analysis are presented in Figure 9, Figure 10, Figure 11, Figure 12 and Figure 13. Additionally, the best examples of each biorecognition technology considering the detection limit are shown in Table 6 and Table 7. The complete table with the 131 articles is presented in Appendix A.

Antibody mimetic molecules technology was, by far, the most used biorecognition technology to detect cancer (Figure 9), representing 93.1% of the studies, while the molecularly imprinted polymers and recombinant antibodies corresponded to 6.1% and 0.8%, respectively. It further supports the previously mentioned answer for RQ2.

According to Figure 10, twelve types of cancer could be diagnosed with non-conventional technologies; the most relevant types are multiple, breast, leukemia, colorectal, and lung (RQ3). The studies focused on biomarkers or cancer entities for detecting multiple types of cancer were the most common, followed by studies about breast. These two categories represented 72% of the analyzed articles. Additionally, the analysis established that a single biomarker or cancer entity could detect different cancer types, as can be observed in the category of multiple types of cancer (RQ4).

The detection methods (transductors) presented in Figure 11 are: electrochemical, optical, magnetic, hybrid (photoelectrochemical, magneto-optical, etc.), microcantilever, and nanopore sequencing (RQ5). The electrochemical method was the most common, followed by the optical method, accounting for 44.3% and 32.8% of the analyzed scientific articles, respectively. Hybrid methods that use magnetic particles to decrease the noise and increase the detection signal were used in 13% of the cases. Two studies utilized microcantilever and nanopore sequencing technologies, which are categorized outside the standard detection methods (electrochemical, optical, and magnetic). However, these technologies are difficult to implement for biosensor development because they require sophisticated equipment.

The most studied biomarkers and cancer entities, presented in Figure 12, were: carcinoembryonic antigen (16.8%), MCF-7 cells (13%), exosomes (12.2%), mucin 1 (10.7%), and human epidermal growth factor receptor 2 (7.6%) (RQ6). It is relevant to point out the increasing interest in non-proteic biomarkers and cancer entities such as cancer cells (22.9%), exosomes (12.2%), and cell-free nucleic acids (4.5%). However, it is relevant to highlight that all the studies for these biomarkers and cancer entities were performed using aptamers, which belong to the antibody mimetic molecules technology. Therefore, there is a lack of studies that use other biorecognition technologies (molecularly imprinted polymers, etc.) for the mentioned type of analysis. Published articles about the three technologies discussed in this review correspond to the best detection limits. These are shown in Table 6, covering the most representative biomarkers and cancer entities per cancer type. In detail, this table shows up to three studies for the cancer types with less than ten articles and seven articles for the rest (breast and multiple), providing 36 studies. Column one provides an identification number for each article to help find the presented studies. Table 6 is ordered by the type of biomarker or cancer entity analyzed to make simpler the study comparisons, seen in column two. The studies were grouped depending on the type of biomarker or cancer entity, in the following order: proteins, nucleic acids, exosomes, and cancer cells. Each group of biomarkers or cancer entities is in alphabetical order, except the exosomes, in which the type of cancer is in alphabetical order. Columns four to seven provide information about the research questions, while columns eight to ten describe the implementation in clinical practice. The references are in column 11. Table 6 and Table 7 are complementary so that the studies shown in each table are different.

Table 7 shows the studies with the lowest limit of detection of the biomarkers studied by at least two of the three biorecognition technologies, allowing an effective comparison of the performance of each technology. This table follows the same structure as the previous table. However, no cancer entity (exosome or cancer cell) is presented because they are studied by only one technology, antibody mimetic molecules.

According to Table 6, most studies used antibody mimetic molecules for biorecognition. Aptamers were the most common molecules for this technology, representing 90% of the total of articles. At least one example for each of the twelve cancer types is shown. Four types of biomarkers and cancer entities were studied: proteins, nucleic acids, exosomes, and cancer cells. At least one study for each type of biomarker or cancer entity type is presented. The most common detection method was the optical (39%), followed by the electrochemical (28%). The non-standard detection methods (microcantilever and nanopore sequencing) are presented for comparison, and they have similar performance to the other methods. The detection limit or lowest analyzed concentration are shown in different units according to the biomarker or cancer entity: proteins and nucleic acids are in molar concentration (from aM to nM), exosomes in particles/mL (from 17 to 10^5^), and cancer cells in cells/mL (from 1 to 213) (RQ7). These ranges of detection are convenient for cancer diagnosis at early stages, especially in early cases when a low detection limit is required. The protein and nucleic acid biomarkers with the lowest detection limit were Mucin 1 (3.3 aM) and microRNAs (5.12 aM), respectively. Both use antibody mimetic molecules and optical detection methods. Additionally, the lowest limits of detection for exosomes and cancer cells were 17 particles/mL (breast cancer exosomes) and one cell/mL (MCF-7 cells and circulating tumor cells). The detection method for these exosomes was electrochemical, while the mentioned cancer cells have studies that used electrochemical, magneto-photoelectrochemical, and nanopore sequencing methods.

On the other hand, 41.7% of the studies presented in Table 6 validated their results successfully in samples of cancer patients, which strongly supports the applicability of these biorecognition technologies in clinical settings. All the real patient samples used in the studies were blood or blood derivatives (plasma or serum). Therefore, the technologies are promising for implementation in blood-based biosensors and tests, but more studies are required to determine their performance in other types of samples or body fluids. Also, it is relevant to mention that the mentioned validations were made with a low number of cancer patients (from 3 to 12), so the statistical significance of these results is limited. For this reason, new validations with a larger number of patients are recommended.

According to Table 7, the biorecognition technology that showed the lowest detection limit for HER2 (0.1 fM) was antibody mimetic molecules with an electrochemical detection method. The detection levels for HER2 biomarker using the three technologies ranged from 0.1 fM to 20 pM, which is smaller than the HER2 blood levels present in cancer patients: 15 to 75 ng/mL (81.1 pM to 0.4 nM) [88]. Therefore, the three technologies presented in the table can be used to diagnose HER2-positive breast cancer. For CA125, the detection level ranged from 0.01 U/mL to 0.015 U/mL, which indicates that antibody mimetic molecules and molecular imprinted polymers have the same performance. The CA125 levels in cancer are usually beyond 35 U/mL [89], so both technologies are adequate for this type of diagnosis. Finally, antibody mimetic molecules technology had a better performance than molecular imprinted polymers in CEA analysis, and the detection levels ranged from 0.66 aM to 0.35 fM. CEA serum levels higher than 5 ng/mL (27.8 pM) can indicate the presence of cancer, so these two technologies can be used in the diagnosis using this biomarker. Three studies (one for CA126 and two for CEA) were validated in serum samples of cancer patients, which represent 42.9% of the table’s articles. As in the previous table, the number of patients used for the clinical validation was very low.

The cancer biomarker market size was approximately USD 11 billion in 2019 and is projected to grow 11.8% each year (up to 2027) [90]. This trend provides a great incentive for implementing the mentioned biorecognition technologies in the development of biosensors. The number of commercially available biosensors that use the mentioned technologies is limited, but there is an excellent perspective to grow in the following years.

Non-conventional biorecognition engineering technologies take advantage of omics technologies to identify biomarkers, cancer entities, molecular signatures, and therapeutic targets involved in this disease. On the order hand, these technologies can be implemented in the construction of biosensors that can diagnose cancer in a low-cost, accessible, and easy-to-implement manner. These interactions are represented in Figure 13. Therefore, in the Venn diagram, it is observed that this systematic review study focused on the three “biorecognition engineering techniques” that are the central parts between the overlapping of cancer studies (orange circle), the biosensors design (green circle), and omics technologies (blue circle). Moreover, the Venn diagram also identifies the other three overlaps between cancer and biosensor design which is the “biomarker analysis,” the overlap of biosensor design and omics technologies which is the “point-of-care analysis,” or the superimposition of omics technologies, and the study of cancer which is the “molecular diagnostics.”

According to the information analyzed in the present article, biorecognition engineering is a promising alternative to conventional antibodies for cancer diagnostics because they overcome some of the limitations that conventional antibodies have, which is especially useful in point-of-care applications. From the mentioned three technologies, the application of antibody mimetic molecules has experienced the most significant increase because this technology can produce highly specific biorecognition molecules at a large scale without requiring cells or organisms. The main limitations of adopting this technology are patents and intellectual property. On the other hand, the molecularly imprinted polymers technology is still reaching its maturity but can represent a good alternative for the mass production of cancer biosensors with high stability. Finally, recombinant antibodies are an alternative to conventional antibody production because they can be produced more efficiently without requiring eukaryotic cells or organisms.

### 4.2. Limitations of the Study

Web of Science was the only database used in this study. In the systematic literature review methodology, the analysis of the scientific studies was conducted only in the window of time of the three most recent years (2019–2021), on papers published in quartile one journals. It only included articles that reported the limit of detection or lowest concentration in the abstract. Due to the above reasons, some relevant articles were expected to be excluded from the analysis. Additionally, the number of reports for the recombinant antibodies and molecularly imprinted polymers may be underrepresented because the ones that do not present a detection concentration in the abstract were excluded.

### 4.3. Future Perspectives

MIP technology presents different advantages over antibodies regarding their stability, ease of preparation, and reduced development costs. Also, they have been used for the development of molecular diagnostic tools and therapeutics that can travel via systemic circulation, find specific target cells, and release drugs into these cells [91]. Despite their great potential, up-to-date MIP technology is still at the initial stages of their development as diagnostic or therapeutic tools against cancer. Currently, there are only a handful of studies about MIP as a drug delivery system for common chemotherapeutic drugs [92]. More investigations in this field are imperative. Additionally, their use as a drug delivery system for chemotherapeutics has been only tested at the analytical phase of development [92]. For that, there is a need to carry out studies about customized MIP-based chemotherapeutic formulations tailored to the specific needs of the drug delivery system to be tested, which should comply with the existing pharmaceutical and biomedical regulations and have minimal safety profile requirements. More studies are needed for MIP to reach maturity and full potential as a diagnostic and therapeutic tool against cancer.

Because of their small size, high stability, strong antigen-binding affinity, deeper tissue penetration, and reduced immunogenicity, nanobodies have great potential to produce recombinant antibodies suitable for development into next-generation molecules for cancer detection and therapeutics. At first glance, one drawback related to the nanobodies’ pharmacokinetics is that, due to their very small size, they can be easily excreted by the kidneys. However, this feature can also become an advantage when they are used for targeted radioimmunoimaging and radioimmunotherapy. In fact, because of their small size, radiolabeled nanobodies are able to cross the blood-brain barrier and are promising vehicles for molecular imaging and targeted radionuclide therapy for metastatic brain lesions [93]. A phase II clinical trial using the 68Ga-NOTA anti-HER2 nanobodies to detect brain metastasis in BC patients is ongoing (ClinicalTrials.gov Identifier: NCT03331601). In addition, the production of diabodies (bispecific antibody fragments that have two antigen-binding domains) by adding to the nanobodies of interest another nanobodies with high binding affinity to human albumin has been performed by Ablynx Company (Ghent, Belgium), and now some nanobodies with this format are in different stages of clinical trials [94]. This can increase the retention time and decrease any possible renal toxicity, allowing better characteristics for the therapeutic application of nanobodies. However, the translation to the clinic is still limited, mainly due to the high cost and long process of establishing good manufacturing practice (GMP) production of nanobodies.

Antibody mimetic molecules, because of their unique properties such as ease of synthesis and modification, high binding affinity, and excellent safety profiles, have been extensively studied for cancer detection and therapeutics, as shown in our analysis as the biorecognition engineering technology with the highest number of the selected publications. Among the most used antibody mimetic molecules, aptamers are the easiest to produce and modify, although affimers and affibodies have higher structural complexities that can increase binding affinity and specificity. However, these last two are more complex to design and modify into an optimal structure. For this reason, affimers and affibodies require more studies and developments.

As reviewed in the initial paragraphs of this section, there is plenty of room for improvement of these three non-conventional technologies against cancer. However, it is important to note that clinical trials and the approval of regulatory entities, such as the Food and Drug Administration (FDA) and European Medicines Agency (EMA), are required to commercialize these applications, either as diagnostic tools or therapeutic drugs. Up to date, no MIP has been approved for cancer treatment. Most of the studies using this technology are at the stage of preclinical evaluation. MIP applications used to deliver the anticancer drugs include 5-Fluorouracil, Doxorubicin, and Paclitaxel [92]. Caplacizumab was one of the first nanobodies (recombinant antibodies) approved in 2019 after clinical trials. It is used to treat thrombotic thrombocytopenic purpura, a blood disorder that causes blood clots. Since then, several nanobodies have been under clinical trials for cancer treatment: anti-HER2 5F7 nanobody, ALX-0141 (target nuclear factor kappa B ligand), DR5Nb1 (target DR5), and ALX-0651 (target CXCR4) [95]. In 2004, the US (FDA) approved the first aptamer drug, Macugen^®^ (pegaptanib), used for the treatment of age-related macular degeneration (AMD). Since then, several aptamers have entered clinical trials to be used as therapeutics for cancer: AS1411 (target nucleolin) and NOX-A12 (target CXCL12) in phase II, and AX102 (target PDGF-B), xPSM-A10 (target PSMA), HB5 (target HER2), HeA2_3 (target HER2), MP7 (target PD-L1), and aptPD-L1 in preclinical trials [96]. On the other hand, current clinical trials for cancer treatment are in progress for several protein scaffolds, of which the following molecules are examples: BMS-986089 (myostatin inhibitor), 18F-BMS986192 (PD-L1 PET tracer), ABY-025 (anti-HER2 affibody), ABY-029 (anti-EGFR affibody), PRS-050 (VEGF-A antagonist), PRS-343 (target HER2), MP0112 (VEGF inhibitor), MP0250 (VEGF and HGF inhibitor), and MP0274 (target HER2) [97]. This information supports the perspective that the mentioned technologies may be adopted in the clinics in the near future.

The computerized molecular design through specialized software can play an essential role in the development and commercialization of the three technologies. The biorecognition design required by them has relied mainly on experiments and directed evolution to produce molecular structures with a high affinity to a specific target. However, advancements in computer-science-generated rational design processes to improve molecular structures have already built and generated massive modifications of these structures towards novel functions [98]. The computerized molecular design has an impact on lowering costs, shortening research periods, optimizing design, increasing reproducibility, facilitating integration with other fields, and improving the understanding of the theoretical bases that provide the basis for molecular structures (chemical bond theory), interactions between molecules (reaction mechanisms) and chemical equilibrium (thermodynamics). For antibody mimetic molecules, one problem present in the interaction with target proteins is that their conformations are rather dynamic, making it challenging to predict mimetic molecule–protein interactions. Thus, major advancements need to occur to improve the prediction of molecular structures to accommodate structural and electrostatic variability in the interaction with the target proteins to generate a wider repertoire of antibody mimetic molecules that can perform several functions and have high specificity of their targets.

## 5. Conclusions

The three objectives presented in the introduction were covered. For the third objective, we highlight a synthesized answer for research questions that gave meaning and relevance to this study. These questions are answered below.
RQ1: Is it possible to implement these technologies to detect cancer?Yes, these technologies can be used for cancer detection. Because of the low levels of detection in some examples, it has been shown to be very promising.RQ2: Which of these three technologies has been more studied?Antibody mimetic molecules were the most used biorecognition technology to detect cancer, comprising 93.1% of the studies. Subsequently, the molecularly imprinted polymers and recombinant antibodies correspond to 6.1% and 0.8%, respectively.RQ3: What types of cancer can be detected using these three technologies?With the study of the systematic review of the literature, we observed that it is possible to detect twelve types of cancer with these technologies. The most relevant types of cancer were multiple, breast, leukemia, colorectal, and lung. However, it is pertinent to highlight that these three technologies may also be efficient in detecting other types of cancer if more studies are conducted in the field.RQ4: Is it possible to detect different cancer types using a single biomarker or cancer entity?Yes, the biomarkers or cancer entities that are used to detect more than one type of cancer are classified into a category called “multiple.”RQ5: What methods are used for the detection of cancer by using these biomarkers?The detection methods are electrochemical, optical, magnetic, hybrid (photoelectrochemical, magneto-optical, etc.), microcantilever, and nanopore sequencing. The most common method was the electrochemical, followed by the optical.RQ6: Which biomarkers and cancer entities are the most commonly studied?The most studied biomarkers and cancer entities were carcinoembryonic antigen (16.8%), MCF-7 cells (13%), exosomes (12.2%), mucin 1 (10.7%), and human epidermal growth factor receptor 2 (7.6%).RQ7: What are the detection levels reached using these three technologies?Biomarkers were detected in concentrations from aM to nM, exosomes in 10^3^–10^5^ particles/mL, and cancer cells in 1–300 cells/mL. Nonetheless, these technologies may be refined in the future to further lower detection levels.Although applications using these technologies are commercially available, there is still a large room for improvements to make them more competitive. Antibodies still dominate the market, but the development of cheap and effective diagnostic devices for cancer is continuously promoting the use of biorecognition engineering technologies. These technologies are emerging tools for developing biosensors and other diagnostic strategies to detect cancer in challenging situations such as the ones found in developing countries and among vulnerable populations.

Finally, with this systematic literature review, we expect that the analysis of the three biorecognition engineering technologies in cancer diagnostics (including their advantages, disadvantages, and perspectives) can help biomedical professionals develop better biosensors for early cancer diagnosis. Moreover, we expect that the material presented in this article can encourage healthcare professionals to use cancer biosensors (as point-of-care diagnostic tools) based on these technologies, which would provide substantial benefits to patients in developing countries and other vulnerable populations.

## Figures and Tables

**Figure 1 cancers-14-01867-f001:**
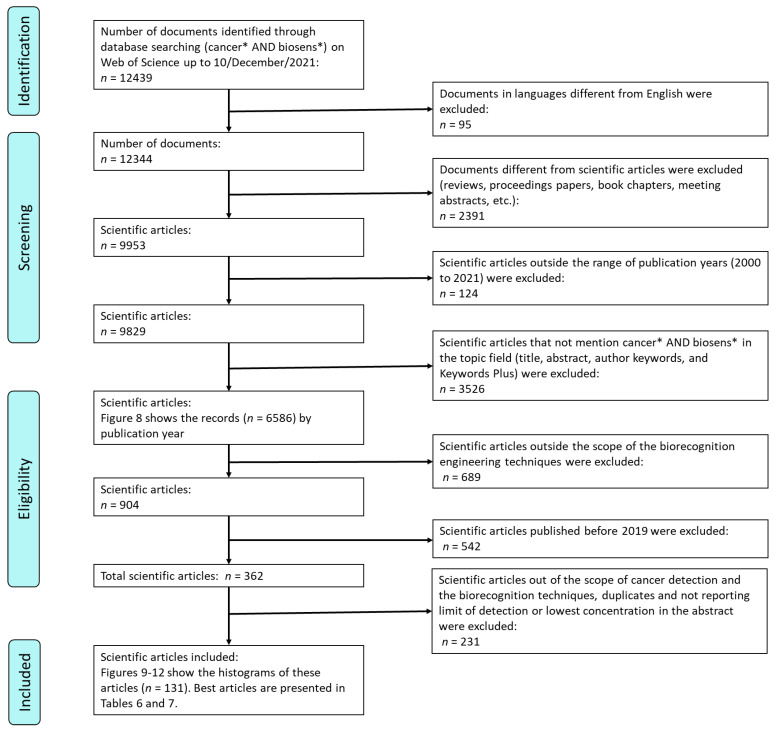
PRISMA flowchart of the article selection process for this study.

**Figure 2 cancers-14-01867-f002:**
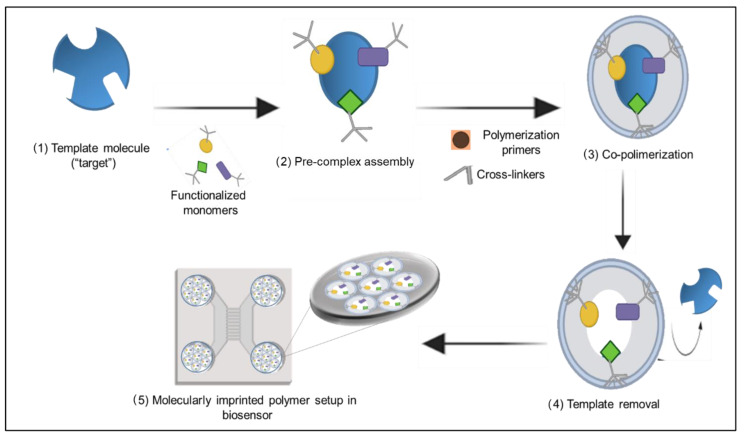
Development process of molecularly imprinted polymers: (**1**) Selection of the target molecule; (**2**) functionalized monomers are added to the target molecule to assemble the pre-complex; (**3**) polymerization primers and cross-linkers are added, and co-polymerization occurs at the established conditions to generate the polymer complex; (**4**) the template is removed using a washing buffer; (**5**) the polymer complex is attached to the surface of the biosensor.

**Figure 3 cancers-14-01867-f003:**
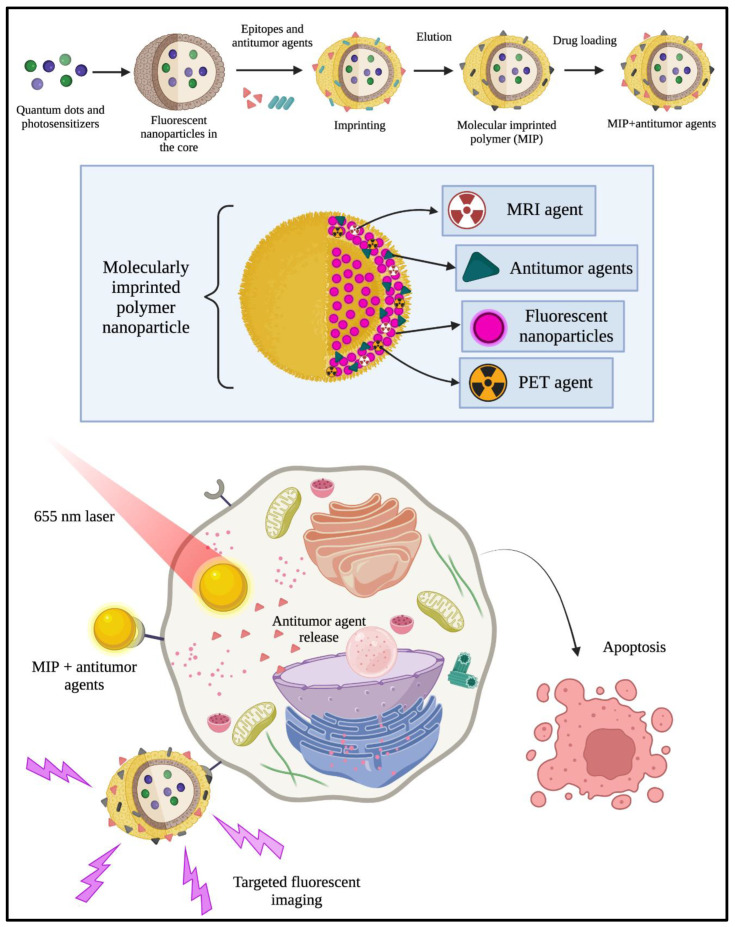
Synthesis of molecularly imprinted polymers encapsulating quantum dots, photosensitizers, MRI contrast agents, and antitumor agents. Such multifunctional loaded MIP particles, when engulfed by the cancer cells and exposed to a laser, leads to activation of apoptosis or programmed cell death and other cascades, finally causing the death of the cancer cells.

**Figure 4 cancers-14-01867-f004:**
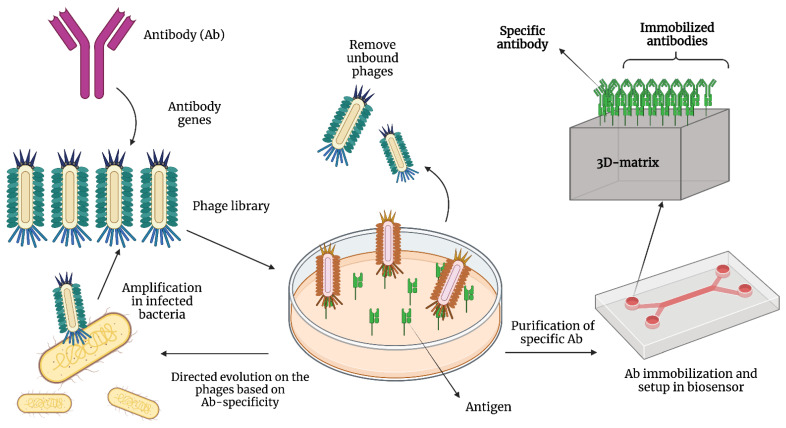
Production of recombinant antibodies using phage display technology. First, a recombinant phage library is created using antibody genes. Next, phages are purified and put in contact with the antigen. Unbound phages are removed. After that, bound phages are eluted and amplified by infecting bacteria. Then, the phages are used to create a new library, and the cycle is repeated until high-affinity antibodies are produced. When that is achieved, the antibodies are purified and immobilized in a biosensor.

**Figure 5 cancers-14-01867-f005:**
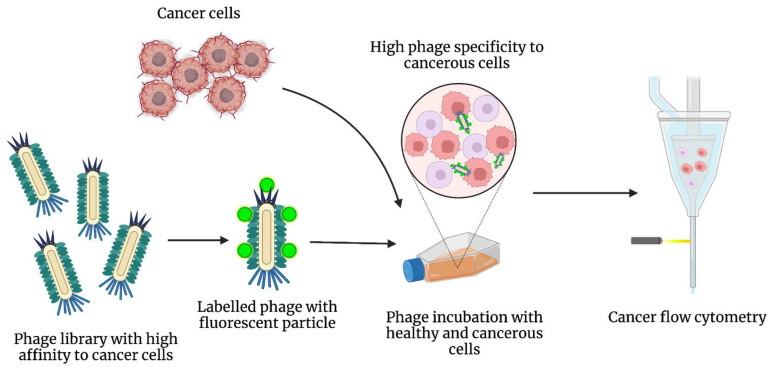
After synthesizing a phage library with a high affinity towards cancer cells, the phages are fluorescently labeled and incubated with the cancer cells. Later, the phage and cancer cell complex can be evaluated in flow cytometry.

**Figure 6 cancers-14-01867-f006:**
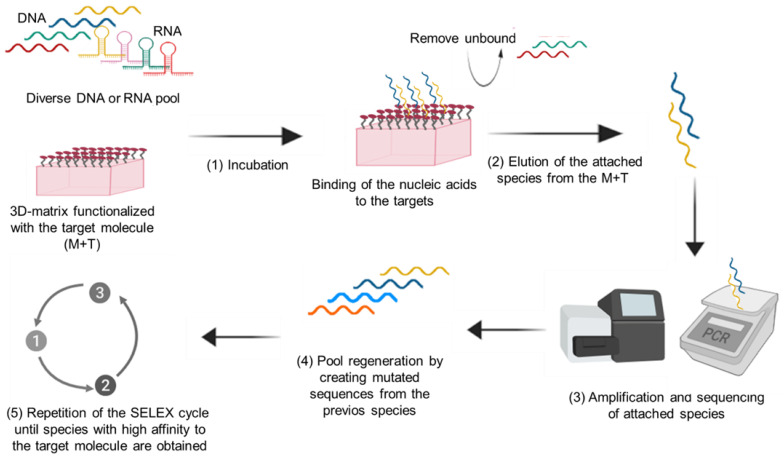
Aptamer development using systematic evolution of ligands by exponential enrichment (SELEX): (**1**) A pool of diverse nucleic acids is incubated with a matrix functionalized with the target molecule; (**2**) unbound nucleic acids are removed while bound nucleic acids (attached species) are eluted; (**3**) the eluted nucleic acids (attached species) are amplified and sequenced; (**4**) the pool is regenerated by creating mutated sequences of the species from the previous step; (**5**) repetition of the SELEX cycle until species with high affinity to the target molecule are obtained.

**Figure 7 cancers-14-01867-f007:**
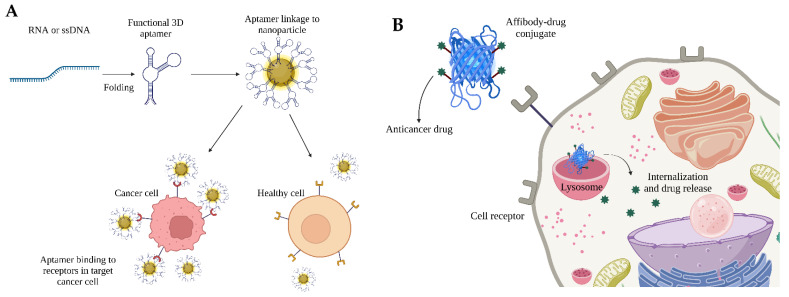
Antibody mimetic molecules are helpful in cancer detection and therapeutics, which can be illustrated as follows: (**A**) Aptamers can be either RNA or ssDNA that can fold itself to form a functional aptamer and can be decorated on the surface of nanoparticles for the specific detection of cancer cells; (**B**) Affibody bound to anticancer drugs can specifically bind cancer cell biomarkers, thus channelizing the drugs inside the cells and aiding drug release leading to cancer cell death.

**Figure 8 cancers-14-01867-f008:**
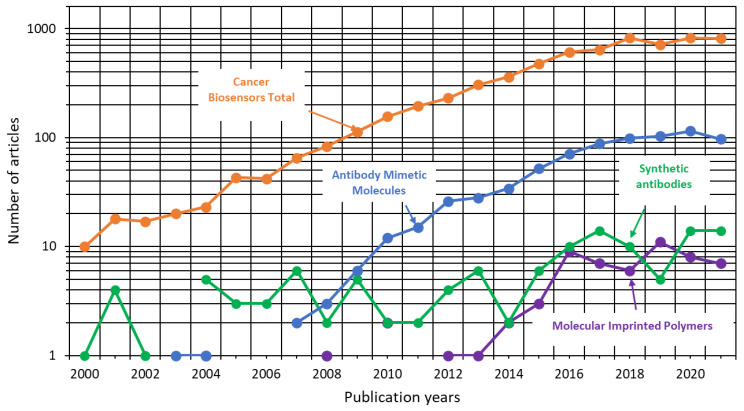
Scientific articles about biosensors and biorecognition engineering technologies for cancer diagnosis published from 2000 to 2021 from Web of Science (up to 10 December 2021).

**Figure 9 cancers-14-01867-f009:**
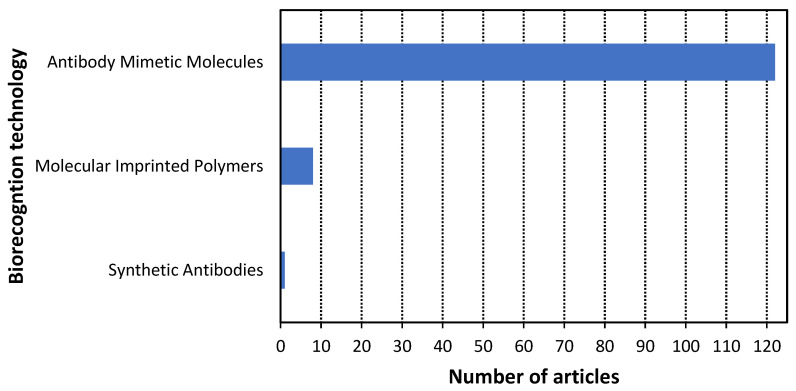
Histogram of the scientific articles about the three biorecognition technologies.

**Figure 10 cancers-14-01867-f010:**
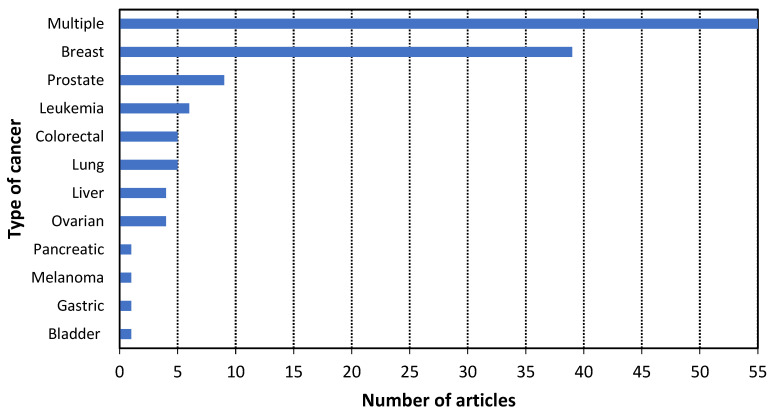
Histogram of the articles of the three biorecognition technologies by type of cancer.

**Figure 11 cancers-14-01867-f011:**
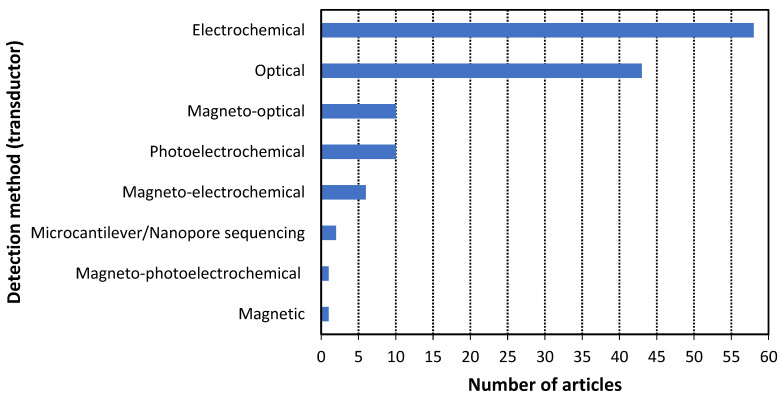
Histogram of the articles of the three biorecognition technologies by detection method.

**Figure 12 cancers-14-01867-f012:**
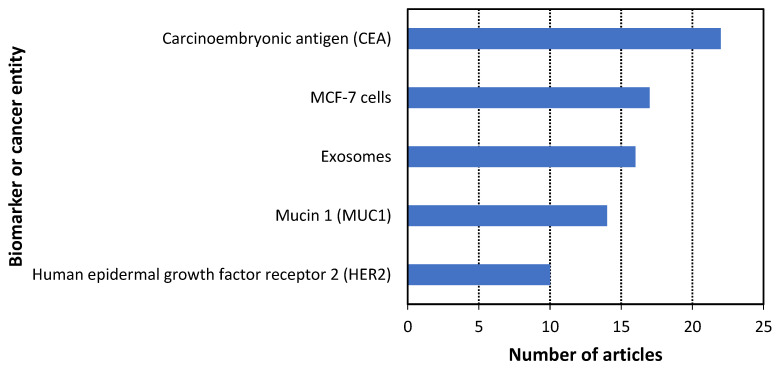
Histogram of the five most studied biomarkers for cancer detection using the three biorecognition technologies.

**Figure 13 cancers-14-01867-f013:**
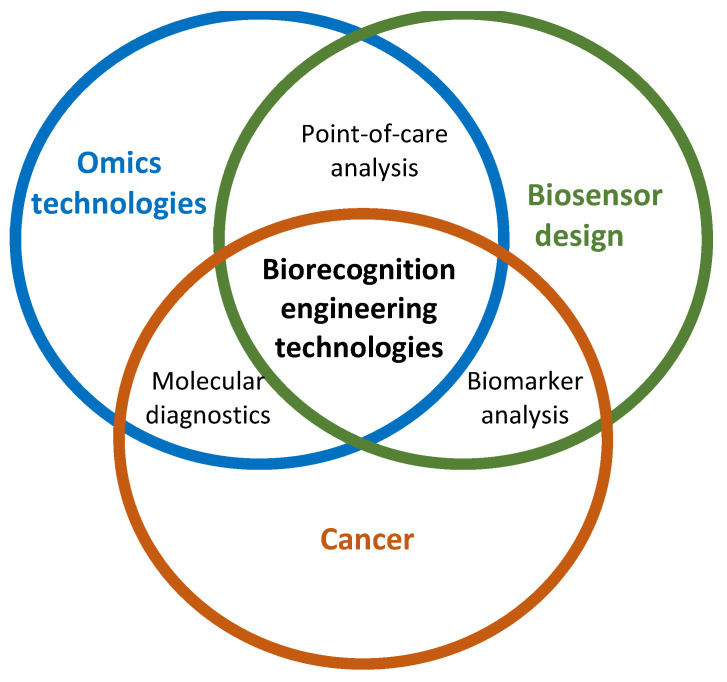
The Venn diagram shows the interactions of the biorecognition engineering technologies with omics technologies, biosensors, and cancer.

**Table 1 cancers-14-01867-t001:** Search methodology used in Web of Science (up to 10 December 2021).

Filter Step on Web of Science	Number Records Obtained
All fields: cancer* and biosens*	12,439
Languages: English	12,344
Document types: Articles	9953
Year published: 2000–2021	9829
Topic: cancer* and biosens*	6586
Topic: molecular* imprint* polymer* OR synthetic antibod* OR phage display OR recombinant antibod* OR aptamer* OR affimer* OR affibody*	904
Publication years: 2019–2021	362
Abstract analysis: duplicates, out of the scope of cancer detection, not reporting limit of detection or lowest concentration, and journal quartile different from 1	131

**Table 2 cancers-14-01867-t002:** Inclusion and exclusion criteria.

Inclusion Criteria	Exclusion Criteria
Scientific articles present in Web of Science	Documents different from scientific articles (reviews, proceedings papers, book chapters, meeting abstracts, etc.)
Articles related to cancer detection	Articles published in languages different from English
Articles related to biosensor development	Articles outside the scope of the biorecognition engineering technologies
Articles published from 2000 to 2021	
Articles published in high impact journals (Quartile 1)	

**Table 3 cancers-14-01867-t003:** Advantages and disadvantages of molecularly imprinted polymers [17,18,19,20].

Advantages	Disadvantages
Its production is cheaper than conventional antibodies because animal or cell lines are not required. It is capable of withstanding harsh conditions: temperatures up to 80 °C and pH range: 2 to 9. It can be easily manufactured in large quantities directly on the devices, so reactors are not needed. Life span at room temperature: 6 to 12 months.	Template leakage. Poor accessibility of the binding sites. Low binding capacity. Non-specific binding. It is difficult to create polymer cavities specific for complex molecules such as proteins. Relatively unstable three-dimensional conformations. Possible rearrangement processes inside the polymer. Poor solubility of the template in solvents, solid substrate.

**Table 4 cancers-14-01867-t004:** Advantages and disadvantages of recombinant antibodies [27,28,29,30].

Advantages	Disadvantages
Its production is less expensive than conventional technologies because it does not require animals or animal cell lines. The structure of the biorecognition element can be manipulated to improve affinity. It can be produced in recombinant bacteria bioreactors with a higher yield than animal cell lines. An increased repertoire of analytes: non-immunogenic small molecules can be analyzed.	It requires library design for recombinant technology. It requires genetic engineering facilities. It contains immunogenic regions that can generate negative immune responses in patients. The use of recombinant technology and protein engineering is usually time-consuming.

**Table 5 cancers-14-01867-t005:** Advantages and disadvantages of antibody mimetics [35,36,37,38,39,40].

Advantages	Disadvantages
Smaller size: more recognition sites in the same surface area increase the sensor’s sensitivity. Libraries with random sequences of DNA, RNA, or amino acids can be used. Directed evolution improves the affinity for the target molecule: KD range 10^−9^ to 10^−12^. The selection of the best candidates can improve thermodynamic and chemical stability. It can use specific functional groups for the attachment on biosensors surfaces. Candidates with reduced immunogenic effects can be selected. It has an extensive repertoire of analytes that are not immunogenic.	It requires library selection and combinatorial mutagenesis. It produces weaker signals than antibodies in some cases. The design and selection methods are complex and time-consuming. In some cases, the use of these technologies is limited by patents and intellectual property rights.

**Table 6 cancers-14-01867-t006:** Applications of biorecognition engineering technologies in cancer biosensors from Web of Science from 2019 to 2021 (up to 10 December 2021).

Article Number	Biomarker or Cancer Entity	Biomarker Abbreviation	Biorecognition Technology	Detection Methods	Limit of Detection/Lowest Concentration Tested *	Cancer Type	Real Patient Samples	Sample Type	Number of Cancer Patients	Reference
1	Alpha-fetoprotein	AFP	Aptamer (AMM)	Magneto-optical	50 pg/mL (0.71 pM)	Liver	No	NA	NA	Wang et al. [46]
2	Alpha-fetoprotein	AFP	Aptamer (AMM)	Electrochemical	60.8 fg/mL (0.87 fM)	Multiple	Yes	Serum	12	Huang et al. [47]
3	Cancer antigen 125	CA125	Aptamer (AMM)	Electrochemical	5.0 pg/mL (45.5 fM)	Ovarian	No	NA	NA	Chen et al. [48]
4	Cancer antigen 125	CA125	Aptamer (AMM)	Electrochemical	0.027 U/mL	Ovarian	Yes	Serum	5	Chen et al. [49]
5	Cancer antigen 125	CA125	Aptamer (AMM)	Magneto-electrochemical	0.08 U/mL	Ovarian	No	NA	NA	Sadasivam et al. [50]
6	Carcinoembryonic antigen	CEA	Aptamer (AMM)	Optical	0.02 pg/mL (0.1 fM)	Lung	No	NA	NA	Shao et al. [51]
7	Carcinoembryonic antigen	CEA	Phage display and Affimer (AMM)	Optical	nM *	Colorectal	No	NA	NA	Shamsuddin et al. [52]
8	Cytochrome C	CYC	Aptamer (AMM)	Optical	1.79 pg/mL (0.15 pM)	Lung	Yes	Serum	NS	Sun et al. [53]
9	Epidermal growth factor receptor and MCF-7 cells	EGFR	Aptamer (AMM)	Electrochemical	5.64 fg/mL (33.2 aM) EGFR and 61 cells/mL MCF-7	Breast	Yes	Serum	NS	Yan et al. [54]
10	Fibroblast growth factor receptor 3	FGFR3	Affimer (AMM)	Electrochemical	pM *	Bladder	No	NA	NA	Thangsunan et al. [55]
11	Human interleukin 2	IL-2	Molecularly imprinted polymer (MIP)	Optical	5.91 fg/mL (0.37 fM)	Multiple	No	NA	NA	Piloto et al. [56]
12	Interleukin 6	IL-6	Aptamer (AMM)	Electrochemical	1.6 pg/mL (76.2 fM)	Colorectal	Yes	Blood	3	Tertis et al. [57]
13	Mucin 1	MUC1	Aptamer (AMM)	Electrochemical	0.72 fg/mL (5 aM)	Multiple	No	NA	NA	Zhao et al. [58]
14	Nuclear ribonucleoprotein A1	HNRNPA1	Affimer (AMM)	Optical	0.1 nM	Colorectal	Yes	Plasma	8	Lee et al. [59]
15	Prostate-specific antigen	PSA	Aptamer (AMM)	Optical	0.54 fM (18.6 fg/mL)	Prostate	No	NA	NA	Chauhan et al. [60]
16	Prostate-specific antigen	PSA	Aptamer (AMM)	Electrochemical	0.043 pg/mL (1.26 fM)	Prostate	No	NA	NA	Chen et al. [61]
17	Vascular endothelial growth factor 165	VEGF165	Aptamer (AMM)	Photoelectrochemical	0.3 fM	Breast	No	NA	NA	Fu et al. [62]
18	Glypican 1 mRNA	GPC1 mRNA	Aptamer (AMM)	Optical	100 fM	Pancreatic	No	NA	NA	Li et al. [63]
19	MicroRNA-21 and Mucin 1	miRNA-21 and MUC1	Aptamer (AMM)	Optical	11 aM miRNA-21 and 0.4 fg/mL (3.3 aM) MUC1	Multiple	No	NA	NA	Li et al. [64]
20	MicroRNAs	miRNAs	Aptamer (AMM)	Optical	5.12 aM	Multiple	No	NA	NA	Zhou et al. [65]
21	Target DNA	NA	Aptamer (AMM)	Magneto-optical	35.5 aM	Lung	No	NA	NA	Zhang et al. [66]
22	Exosomes	NA	Aptamer (AMM)	Electrochemical	17 particles/mL	Breast	Yes	Plasma	NS	Kashefi-Kheyrabadi et al. [20]
23	Exosomes	NA	Aptamer (AMM)	Optical	4.27 × 10^4^ particles/mL	Gastric	Yes	Plasma	NS	Huang et al. [67]
24	Exosomes	NA	Aptamer (AMM)	Electrochemical	920 particles/μL (92 × 10^3^ particles/mL)	Leukemia	No	NA	NA	Yu et al. [68]
25	Exosomes	NA	Aptamer (AMM)	Electrochemical	84 particles/μL (8.4 × 10^3^ particles/mL)	Liver	Yes	Blood	10	Wu et al. [69]
26	Exosomes	NA	Aptamer (AMM)	Optical	2.5 × 10^3^ particles/mL	Multiple	Yes	Serum	12	Liu et al. [70]
27	Exosomes	NA	Aptamer (AMM)	Optical	1 × 10^5^ particles/mL	Prostate	Yes	Serum	10	Chen et al. [71]
28	B16-F10 cells	NA	Aptamer (AMM)	Electrochemical	33 cells/mL	Melanoma	No	NA	NA	Liu et al. [72]
29	CCRF-CEM and MCF-7 cells	NA	Aptamer (AMM)	Photoelectrochemical	5 cells/mL CCRF-CEM and 10 cells/mL MCF-7	Multiple	No	NA	NA	Wang et al. [73]
30	Circulating tumor cells	NA	Aptamer (AMM)	Magneto-optical	>1 cell/mL	Liver	Yes	Blood	NS	Gopinathan et al. [74]
31	Circulating tumor cells	NA	Aptamer (AMM)	Nanopore sequencing	1 cell/mL	Breast	Yes	Blood	7	Li et al. [75]
32	K562 cells	NA	Aptamer (AMM)	Electrochemical	60 cells/mL	Leukemia	No	NA	NA	Zheng et al. [76]
33	MCF-7 cells	NA	Aptamer (AMM)	Electrochemical	1 cell/mL	Breast	Yes	Blood	8	Shen et al. [77]
34	MCF-7 cells	NA	Aptamer (AMM)	Magneto-photoelectrochemical	1 cell/mL	Breast	No	NA	NA	Luo et al. [78]
35	MCF-7 cells and Mucin 1	MUC1	Aptamer (AMM)	Microcantilever	213 cells/mL and 0.9 nM MUC1	Breast	No	NA	NA	Li et al. [79]
36	Ramos and CCRF-CEM cells	NA	Aptamer (AMM)	Magneto-electrochemical	4 cells/mL Ramos and 3 cells/mL CCRF-CEM	Leukemia	Yes	Blood	NS	Dou et al. [80]

* Note: ‘*Limit of detection*’ refers to the concentration value detected when a calibration curve was made (referring to a linear behavior). On the other hand, ‘*lowest concentration tested*’ refers to the minimum concentration value tested and reported in the experiments carried out in the study. MIP: molecularly imprinted polymers, RA: recombinant antibodies, AMM: antibody mimetic molecules, NA: not applicable, NS: not specified.

**Table 7 cancers-14-01867-t007:** Comparison of three biorecognition technologies using different cancer biomarkers.

Article Number	Biomarker or Cancer Entity	Biomarker Abbreviation	Biorecognition Technology	Detection Methods	Limit of Detection/Lowest Concentration Tested *	Cancer Type	Real Patient Samples	Sample Type	Number of Cancer Patients	Reference
1	Carbohydrate antigen 125	CA125	Molecularly imprinted polymer (MIP)	Electrochemical	0.01 U/mL	Ovarian	No	NA	NA	Rebelo et al. [81]
2	Carbohydrate antigen 125	CA125	Aptamer (AMM)	Optical	0.015 U/mL CA125	Multiple	Yes	Serum	2	Xu et al. [82]
3	Carcinoembryonic antigen	CEA	Molecularly imprinted polymer (MIP)	Magneto-optical	0.064 pg/mL (0.35 fM)	Multiple	Yes	Serum	3	Lin et al. [83]
4	Carcinoembryonic antigen	CEA	Aptamer (AMM)	Photoelectrochemical	0.12 fg/mL (0.66 aM)	Multiple	Yes	Serum	NS	Gao et al. [84]
5	Human epidermal growth factor receptor 2	HER2	Recombinant antibody (RA)	Optical	20 pM	Breast	No	NA	NA	Dong et al. [85]
6	Human epidermal growth factor receptor 2	HER2	Molecularly imprinted polymer (MIP)	Electrochemical	0.43 ng/mL (2.3 pM)	Breast	No	NA	NA	Lahcen et al. [86]
7	Human epidermal growth factor receptor 2 and MCF-7 cells	HER2	Aptamer (AMM)	Electrochemical	19 fg/mL (0.1 fM) HER2 and 23 cells/mL	Breast	No	NA	NA	Gu et al. [87]

* Note: ‘*Limit of detection*’ refers to the concentration value detected when a calibration curve was made (referring to a linear behavior). On the other hand, ‘*lowest concentration tested*’ refers to the minimum concentration value tested and reported in the experiments carried out in the study. MIP: molecularly imprinted polymers, RA: recombinant antibodies, AMM: antibody mimetic molecules, NA: not applicable, NS: not specified.

## Data Availability

Data from Clarivate Web of Science was analyzed in this study. This data is available in (subscription required): https://webofknowledge.com (accessed on 14 March 2022). In addition, in Appendix A in this document, it is possible to observe the 131 studies taken for this systematic literature review, link: https://docs.google.com/spreadsheets/d/1ETKQOOO0kNlptgM6nwf38rS44OzAupF4/edit?usp=sharing&ouid=116879916920386330424&rtpof=true&sd=true (accessed on 14 March 2022) or in this repository: https://hdl.handle.net/11285/645288 (accessed on 14 March 2022).

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
