# Peer review of "Biorecognition Engineering Technologies for Cancer Diagnosis: A Systematic Literature Review of Non-Conventional and Plausible Sensor Development Methods"

_cancers, 2022, doi:10.3390/cancers14081867_

Round 1
Reviewer 1 Report
The review paper covers the studies conducted on three general subjects, namely, molecularly imprinted polymers, sythetic antibodies and mimetic antibodies. I think the coverage of the non-traditiobal recognition systems are good enough for a broad range of audience from several disciplines.
I have a few comments for the authors, which could improve the final quality of the paper.
- Figure1: This figure is too simple to be a first figure in such a review paper. The authors are encouraged to prepare a combined image of the affinity agents along with their application field. The figure is far from sophistication in its current form.
- Table 6: A new column should be added and the articles that used real patient samples for the detection should be highligted there.
- Table 6: The number os the studies presented in the table should be doubled wherever possible.
- Table 7 should be extended for comparison of other biomarkes (other than HER2) with the defined affinit technologies.
- Figure 14 does not provide any specified information and should be improved with more parameters/data/etc.
Reviewer 2 Report
In the review manuscript by Mayoral-Peña et al., the authors provide an intensive revision about “Biorecognition engineering technologies for cancer diagnosis: A systematic literature review of non-conventional and plausible sensor development methods”
The topic of the manuscript and the articles reviewed are appropriate for the journal. Articles are mostly from recent years.
However, there are major and minor points that need to be addressed prior to publication.
- In the opinion of the reviewer, a biomarker is just a naturally occurring molecule, gene, or characteristic associated to a particular physiological or pathological process, disease, etc.; and thus, MCF-7 cells and exosomes should not be considered as biomarkers. The authors should modify the whole manuscript regarding this point. Maybe, the authors could referred as cancer entities for exosomes and MCF-7 (or other cancer cells) cells.
- In figures 1 and 5 the authors are referring to Phage Display. However, the phage depicted in the figures is a lytic one (T7 phage) mostly used for peptide phage display and not for antibody phage display. Antibody phage display is mainly performed using M13 phages (or derived phages). Therefore, the authors should alternatively modify the figures or acknowledge in the legend to the figures the phage used for that kind of applications.
- Synthetic antibodies along the manuscript. The authors referred to synthetic antibodies, however according to their production they should be referred as recombinant antibodies. Indeed, “Synthetic” or “semi-synthetic” are terms mostly associated to the antibody libraries, depending on how they were constructed (i.e. by introducing the diversity by means of synthetic degenerated oligonucleotides). Please, correct along the manuscript.
Minor points:
- Figure 2 and Table 1 are redundant. The information depicted is the same.
- Lines 340-343. The synthetic peptide should be an example of peptide phage display not about synthetic antibody technology as is indicated in the paragraph. Please carefully edit the manuscript for these inconsistencies.
- Figure 13. Modify the figure to not indicate that exosomes and MCF-7 cells are cancer biomarkers
- Table 6. Column limit of detection/… What does* mean? Please indicate at the legend to the table.
Full column, it seems that all units are not properly indicated. Please fix. (i.e. Pm, should read pM, Fm to fM … ¿?)
- Lines 544-545, lines 554-557, RQ6 and along the manuscript. A non-proteic biomarker should be a genetic biomarker, lipid, metabolite or another unique molecule but no cancer entities as cancer cells or exosomes derived from cancer cells or tumoral cells. The authors should edit that sentence and along the manuscript to avoid for inconsistencies.
Typos:
consists in to consists of (line 202)
Table 4: production is less expensive comparison. Please fix
VH and VL should be represented as VH and VL
line 397 are are. Fix.
Table 5. Sensibility in Advantages should read sensitivity
Figure 9 to 12 are all necessary? Some for supporting information?
Lines 560. It seem there is a problem with the units. Nm and Am should be nM and aM.
ensibility in Advantages should read sensitivity
Figure 9 to 12 are all necessary? Some for supporting information?
Lines 560. It seem there is a problem with the units. Nm and Am should be nM and aM.
Author Response
"Please see the attachment."

Round 2
Reviewer 2 Report
The revised version of the manuscript has been considerably improved.
The authors have properly addressed previous comments and concerns, therefore the manuscript is now recommended for publication.